# METER (Mental health emergency response) program: Findings of psychological impact status and factors associated with depression, anxiety and stress among healthcare workers in public hospital in Malaysia during the COVID-19 pandemic

Nor Asiah Muhamad[1]☯*, Natasha Subhas[2]☯, Normi Mustapha[3], Norni Abdullah[2], Muhammad Arif Muhamad Rasat[2], Rimah Melati AB Ghani[1], Fatin Athira Tahir[4], Anne Nik Ismaliza Ishak[2], Vevehkanandar Sivasubramaniam[2], Alinazarine Hassan[2], William Wei Liang Goh[2], Kok Liang Teng[2], Ainul Izzah Abdul Manan[2], Rosmawati Mokhtar[2], Amrit Kaur Baljit Singh[2], Kher Shean Ng[2]

**1** Sector for Evidence Based Healthcare, National Institutes of Health, Ministry of Health, Shah Alam, Selangor, Malaysia, **2** Department of Psychiatry and Mental Health, Hospital Tengku Ampuan Rahimah, Ministry of Health, Klang, Malaysia, **3** Faculty of Science & Technology, Open University Malaysia, Petaling Jaya, Malaysia, **4** Institute for Public Health, National Institutes of Health, Ministry of Health, Shah Alam, Malaysia

☯ These authors contributed equally to this work.
* norasiahdr@gmail.com

## Abstract

### Introduction

The COVID-19 pandemic has become the greatest challenge of the new millennium. Most healthcare workers (HCWs) experienced unprecedented levels of workload since the pandemic. This study aims to identify the prevalence and factors of depression, anxiety and stress among HCWs in Malaysian healthcare facilities in the midst of the pandemic due to the SARs-CoV-2.

### Methods

An emergency response programme on mental health was conducted from June to September 2020. A standardized data collection form was distributed among the HCWs in the government hospital in Klang Valley. The form contained basic demographic information and the self-reported Malay version of the Depression, Anxiety and Stress scale (BM DASS-21).

### Results

Of the1,300 staff who attended the Mental Health and Psychosocial Support in Covid-19 (MHPSS COVID-19) programme, 996 staff (21.6% male, 78.4% female) completed the online survey (response rate: 76.6%). Result showed that staff aged above 40 years old were almost two times more likely to have anxiety (AOR = 1.632; 95% CI = 1.141–2.334,

**Data Availability Statement:** The original contributions presented in the study are included in the article. Further inquiries can be directed to the corresponding author.

**Funding:** This METER program received no specific grant from any funding agency in public, commercial, or not-for-profit sectors. This publication is funded by the National Institutes of Health (NIH), Ministry of Health, Malaysia with NMRR registration ID: NMRR ID-21-02186-QTG.

**Competing interests:** The authors have declared that no competing interests exist.

p:0.007) and depression (AOR = 1.637; 95% CI = 1.1.06–2.423, p:0.014) as compared to staff who were less than 40 years old. Those who had direct involvement with COVID-19 patients were likely to suffer stress (AOR = 0.596; 95% CI = 0.418–0.849, p:0.004), anxiety (AOR = 0.706; 95% Ci = 0.503–0.990, p:0.044) and depression (AOR = 0.630; 95% Ci = 0.427–0.928, p:0.019). HCWs with stress (AOR = 0.638; 95% CI of 0.476–0.856, p = 0.003), anxiety (AOR = 0.720; 95% CI 0.542–0.958, p = 0.024) and depression (AOR = 0.657; 95% CI 0.480–0.901, p = 0.009) showed less confidence to treat critically ill patients and need psychological help during outbreak.

## Conclusion

This study showed the importance of psychosocial support to reduce psychological distress among HCWs when working or coping during the COVID-19 pandemic or outbreak.

## 1. Introduction

The coronavirus disease 2019 (COVID-19) first emerged in December 2019 in Wuhan City, China following reports of several pneumonia cases with unknown aetiology agent [1]. The aetiological agent was soon identified as severe acute respiratory syndrome coronavirus 2 (SARS-CoV-2) [2, 3]. COVID-19 was considered as a Public Health Emergency of International Concern (PHEIC) by World Health Organization (WHO) due to its rapid spread and declared a pandemic on 12 March [4–6].

Malaysia and other Southeast Asian countries were among the first few countries to report incidences of the virus outside China [7, 8]. The first confirmed case of COVID-19 in Malaysia was detected among three Chinese nationals on 25 January 2020 who previously had close contact with people infected with COVID-19 in Singapore [7, 8]. However, following the rapid increase in active COVID-19 cases, the Malaysian Government ordered the Movement Control Order (MCO) beginning on 18 March 2020 [9]. The MCO was enforced as a mitigation effort to reduce community spread and decrease the overburdening of the country's health system [10].

Psychological distress during a pandemic is common, be it to the general public or health care workers. Various studies have reported the consequences of the first massive disease outbreak of the 21st century, severe acute respiratory syndrome (SARS) [11]. These studies found that in the immediate aftermath of the SARS outbreak, psychiatric problems commonly diagnosed were adjustment reactions, stress and increased anxiety levels [11–16]. A study conducted during the SARS outbreak in a large teaching hospital in Toronto reported almost two-thirds of healthcare workers (HCWs) surveyed experienced high levels of psychiatric distress [17]. Working during or immediately after an outbreak of an infectious disease or pandemic which may affect the healthcare system, will have a negative impact on the mental health and well-being of the HCWs.

Various factors will affect the mental health and well-being of the HCWs during a pandemic such as concern about exposure to the virus; personal and family needs and responsibilities; managing a different workload; lack of access to necessary tools and equipment (including personal protection equipment, PPE), learning new technical skills and adapting to a different work [18]. Six months' post-discharge, major depression (23.6%), adjustment disorder (8.1%) and post-traumatic stress disorder (PTSD) (7.3%) were common psychopathologies seen in SARS patients [19]. A study by Mak et al., (2009) reported high incidences of mental health disorders at 58.9% following the SARS outbreak [11].

As mentioned above, common psychological sequelae seen during and after a pandemic are stress, depression and anxiety among HCWs as well as the general population [11, 20, 21]. Recent studies like Wang C et al. [22] and Qiu J et al., [23] found that their community suffered from severe psychological distress (stress, anxiety and depression) during the COVID-19 pandemic. From experience, stress, depression and anxiety are not only common amongst the general population, the prevalence and risk are significantly higher for the high-risks populations such as healthcare workers [24, 25].

Healthcare workers not only engage in the treatment of infected patients, but are also at risk of being infected themselves. Apart from the risk of being of infected, they face additional stressors, such as the fear of transmitting COVID-19 to family members, being stigmatized or rejected by others based on their occupation, and working under highly stressful conditions [26]. Over time, the increasing COVID-19 cases with associated deaths, a heavy workload for long periods and limited availability of personnel protective equipment could further cause physical and emotional burnout in these healthcare workers [26]. There are several studies conducted on healthcare workers' mental health during the COVID-19 pandemic. Pappa S et al. [27] reported that one in every five healthcare workers suffers from depression, anxiety or both. Another review concluded that healthcare workers reported more anxiety, depression and sleep problems [28]. Badahdah A et al. [29]'s study in Oman found that there was a high prevalence of stress, anxiety and poor psychological well-being amongst young health care workers during the COVID-19 pandemic. Studies from China found that frontline HCWs in direct contact with COVID-19 patients suffered stress, anxiety and insomnia and higher levels of psychiatric symptoms compared to those with indirect contact [30, 31]. A large, retrospective cohort study involving 62,354 participants in the United States by Taquet et al., 2021 reported survivors of COVID-19 appear to be at increased risk of psychiatric sequelae, and a psychiatric diagnosis might be an independent risk factor for COVID-19 [32]. Therefore, it is pertinent to place importance on early detection and management of psychological sequelae in case of future pandemic outbreaks to avoid a 'Mental Health Catastrophe' [11, 31].

COVID-19 is itself a huge stressor, as there were no existing medicine or vaccines during the start of the pandemic [33]. Stress can be defined as a 'state of disharmony and is neutralized by a complicated variety of physiologic and behavioural responses that aim to maintain/re-establish the threatened homeostasis (adaptive stress response)' [34, 35]. Stress has a wide spectrum of symptoms ranging from emotional to physical disorders like sadness, anxiety, palpitations, and gastrointestinal distress. The higher the stress, the more its symptoms like frequent headaches, fatigue, neck/back pain, excessive worries, muscle tension and feeling overwhelmed [34]. Many people experience poor concentration, forgetfulness and low energy as a stress response to the pandemic [34]. Depression is a common but complex disorder with a range of unique symptomology that includes persistent low mood, anhedonia, poor concentration, sleep disturbances, fatigue and in more severe forms lead to impairment in function and suicidal ideations [36, 37]. Anxiety disorder is an anxiety disorder characterised by excessive overthinking and worry, feeling at the edge, muscle tension, poor concentration associated with symptoms of hypervigilance, and other somatic symptoms of anxiety [36, 38]. Another common anxiety disorder, panic disorder, is characterised as sudden, sometimes unexpected paroxysmal bursts of severe anxiety usually associated with several physical symptoms (autonomic, otoneurological, cardiorespiratory or gastrointestinal) which can be disabling and affect a person's function [36, 39]. The Depressive, Anxiety and Stress Scale (DASS) was developed specifically to measure depression, anxiety and stress levels concurrently [40, 41]. Each of the three subscales of DASS is intercorrelated with another [40, 41]. During the COVID-19 coronavirus pandemic WHO warned about the potential negative impact of the crisis on the psychological and mental well-being of the population particularly health and social care

professionals [42]. Thus, evaluating how individuals perceive stressful situations in their lives is critical for the quantification of psychological stress among HCWs.

An increase in psychological distress, anxiety and depression showed a need for a programme to support psychologically the HCWS and other professionals [18, 23, 29]. Developing and promoting resilience should protect people from stress and psychopathological symptoms during the COVID-19 outbreak [37]. Even though perseverance and resilience are associated with mental health outcomes, no data are available for HCWs in Malaysia. Therefore, this study aimed to determine the prevalence of depression, anxiety, and stress among HCWs in public hospitals in Malaysia. This study hypothesised a relevant prevalence rate for moderate to severe psychological distress among HCWs. The negative impact on health and social care professions may result in effects at multiple levels, from the individual worker to the entire health and social care system at the macro level. The finding from this study is useful to assess the health and well-being of HCWs during the COVID-19 pandemic and provide psychological support to HCWs for the sustainability and functionality of the workforce and healthcare system.

## 2. Methods

### Ethics statement

This study approval was obtained from the National Medical Ethics Committee with NMRR ID-21-02186-QTG and Medical Research Ethics Committee at the National Institutes of Health, Malaysia. Subjects of this study consented to participation in this study before data collection.

### Study design, participant and setting

During the movement control order, a serial programme on Mental Health and Psychosocial Support in Covid-19 (MHPSS COVID-19) was activated in all tertiary hospitals in Klang Valley, Malaysia throughout 2020. The serial programme on MHPSS COVID-19 was planned for all HCWs in various clinical fields managing COVID-19 patients at Hospital. The serial programme on MHPSS COVID-19 consists of education on stress management, relaxation technique, and hotline access for early psychological and emotional support. The MHPSS COVID-19 providers consisted of psychiatrists, medical officers, clinical psychologists and counsellors who were tasked to assess the mental health status of all the HCWs of their hospitals.

A cross-sectional questionnaire-based survey was conducted throughout 2020. HCWs working at the government hospital within the Ministry of Health, Malaysia who attended the MHPSSS COVID -19 were eligible to participate in the survey. Exclusion criteria included those under 18 years of age, non-Malay speakers, and non-HCWs. The study participants were recruited during this programme. Those who agreed to participate were subsequently asked to respond to the Malay version of the Depression, Anxiety and Stress scale (Malay Version of DASS-21). The assessment was conducted through a google form and subsequently, Psychological First Aid (PFA) and Virtual Psychological Intervention (Please see the description in the sub-section) were provided to the HCWs who had psychological distress.

### Data collection

Upon recruitment, a Psychological First Aid (PFA) form was distributed online as a first aid screening which was sent out via QR code and electronic messaging system to all participants in the programme. Subsequently the Virtual Psychological Intervention were provided to the participants who had psychological distress beside the Psychological First Aid (PFA).

Participants who agreed to participate were subsequently asked to respond to the Malay Version of DASS-21 via a google form. We adopted the modified version of the Depression, Anxiety, and Stress Scale–21 (Malay Version of DASS-21), which is a reliable and valid self-administered instrument to screen for these psychological disorders. The Malay version of the DASS 21 item is a modified version of the original 42 items (DASS 42) and is a self-reported instrument requiring no special skills to administer. The malay version of the DASS-21 has good psychometric properties and is culturally suitable to be used for the Malaysian general population [40, 41].

## Psychological first aid screening application and reporting

All participants in the programme received a Psychological First Aid (PFA) screening form online via QR code scanning or an electronic messaging system. The PFA form contained two main sections. The first section contains information on socio-demographic such as gender, current age, ethnicity, nationality, profession, and involvement with COVID-19 patients. The second section contains information on the Malay version of the Depression, Anxiety and Stress scale (BM DASS-21). The DASS-21 scoring was coded as normal, mild, moderate, and severe. All HCWs received their results upon completion of the PFA screening. Any HCWs with a normal score received a phone call or electronic message after two weeks. For HCWs who scored 'mild/moderate' were requested to attend a teleconsultation session by the counsellor with 'Self-help' materials. Other HCWs with a 'severe' score were asked to attend a teleconsultation session by the clinical psychologist or medical officer or psychiatrist (Fig 1).

## Statistical analysis

Univariate, bivariate, and multivariate statistical analyses were conducted. Frequency and percentages were used to describe characteristics and estimate prevalence rates of depression, anxiety, and stress among participants. A bivariate analysis was conducted using Pearson's chi-square test to explore the association between sociodemographic traits and psychological characteristics and support with each DASS subscale. Variables that were significantly (p-value is considered to be less than 0.05) associated with the outcomes were further analysed by entering the adjusted multivariate model. Multivariate logistic regression analysis determined the factors associated with each outcome (i.e., depression, anxiety, and stress). Adjusted odds ratios with 95% confidence intervals (CIs) and p-values were calculated to determine the strength and significance of the association. All statistical analyses were performed using SPSS, version 22.0 (SPSS Inc., Chicago, Ill., USA).

## 3. Results

### Participants' characteristics

A total of 1300 staff participated in the MHPSS COVID-19 programme. Of these, 996 completed the online survey which gave a response rate of 76.6%. Of the total participants, 78.4% (781) were female, 80.2% (799) were below 40 years of age and 68.7% (684) were support staff (Table 1). Table 2 showed the distribution of depression, anxiety and stress among health workers.

With regards to the relaxation tips being provided by the hospital's psychological response team in the google form, 955 staff (95.9%) found it useful. The survey revealed 892 staff (89.6%) were worried about contracting COVID-19 and 942 (94.6%) were worried they may spread it to their family members. Although 903 staff (90.7%) were worried about the shortage

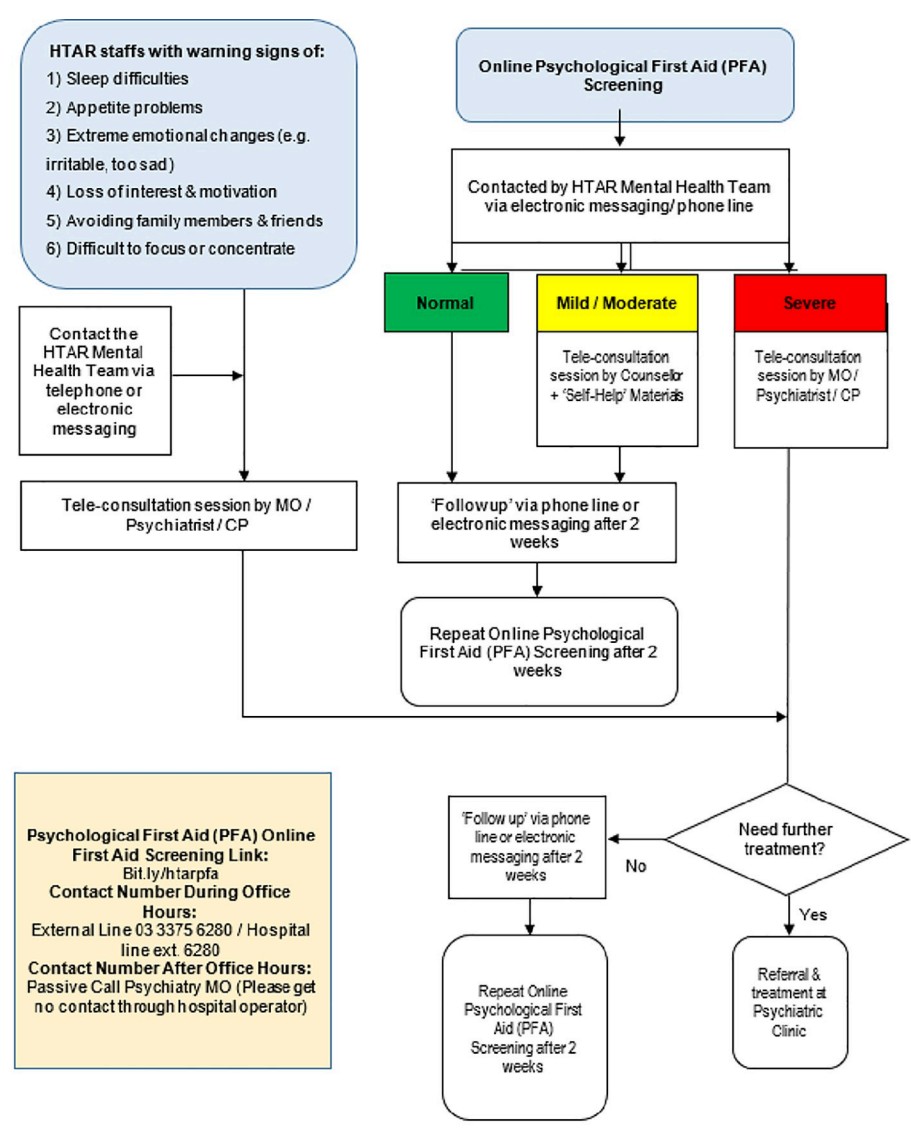

*MO = Medical Officer / CP = Clinical Psychologist*

**Fig 1. Flow chart of Psychological First Aid (PFA), Mental Health Team, Hospital Tengku Ampuan Rahimah (HTAR).** *MO = Medical Officer / CP = Clinical Psychologist.

of Personal protective equipment, 514 staff (51.6%) were quite confident about treating critical patients.

A total of 493 (49.5%) of the staff felt that adequate working hours during the pandemic should be between 6 to 8 hours, followed by 4 to 6 hours (28.8% or 267) and 8 to 10 hours (124 or 12.4%). During the pandemic, 733 (73.6%) of the staff felt that they had time to rest physically and mentally. A total of 866 staff (86.9%) were satisfied with how the hospital was supporting them during the pandemic and 995 (95.6%) of the staff declined psychological intervention provided by the hospital. The staff were asked to fill up the online DASS-21 scoring, a tool used to measure psychological distress (Table 3).

**Table 1. Distribution of sociodemographic characteristics of the healthcare workers (N = 996).**

| Characteristic/ Variable | frequency (%) |
|---|---|
| Gender | |
| Male | 215 (21.6) |
| Female | 781 (78.4) |
| Job Category | |
| Support Staff | 684 (68.7) |
| Technical Staff | 312 (31.3) |
| Age Group | |
| < 40 years old | 799 (80.2) |
| ≥ 40 years old | 197 (19.8) |

## Depression

With regards to depression, this study (Table 2) showed that about 77.8% (775) were normal, 10.4% (104) were mild, 7.1% (71) were moderate, and 4.6% (46) were severe. Therefore, 77.8% (775) were normal and 22.2% (221) had depression. Staff aged below 40 years old were twice more likely to have depression as compared to those aged 40 and above (AOR = 1.637; 95% CI: 1.106–2.423, p = 0.014) (Table 4). Staff with no involvement with COVID-19 patients are less likely to have depression (p = 0.011) as compared to those who have direct involvement with COVID-19 patients (AOR = 0.630; 95% CI:0.427–0.928, p = 0.019). Staff who have no time to rest during the COVID-19 pandemic are twice more likely to have depression as compared to

**Table 2. Distribution of depression, anxiety and stress among healthcare workers (N = 996).**

| Characteristic/ Variable | frequency (%) |
|---|---|
| Depression Category | |
| Normal | 775 (77.8) |
| Mild | 104 (10.4) |
| Moderate | 71 (7.1) |
| Severe | 46 (4.6) |
| Depression | |
| No | 775 (77.8) |
| Yes | 221 (22.2) |
| Anxiety Category | |
| Normal | 716 (71.9) |
| Mild | 71 (7.1) |
| Moderate | 125 (12.6) |
| Severe | 84 (8.4) |
| Anxiety | |
| No | 716 (71.9) |
| Yes | 280 (28.1) |
| Stress Category | |
| Normal | 727 (73.0) |
| Mild | 186 (18.7) |
| Moderate | 61 (6.1) |
| Severe | 22 (2.2) |
| Stress | |
| No | 727 (73.0) |
| Yes | 269 (27.0) |

**Table 3. Distribution of psychological characteristics among healthcare workers (N = 996).**

| Characteristic/ Variable | frequency (%) |
|---|---|
| Involvement with Covid 19 patients | |
| No | 320 (32.1) |
| Direct | 275 (27.6) |
| Indirect | 401 (40.3) |
| Do you think these tips are useful to you and can be practiced in the current situation? | |
| No | 41 (4.1) |
| Yes | 955 (95.9) |
| Are you worried that you will be infected with Covid-19? | |
| No | 104 (10.4) |
| Yes | 892 (89.6) |
| Are you afraid that you could infect your family? | |
| No | 54 (5.4) |
| Yes | 942 (94.6) |
| Are you worried about the lack of PPE (Personal protective equipment)? | |
| No | 93 (9.3) |
| Yes | 903 (90.7) |
| Do you feel less confident to treat critically ill patients? | |
| No | 514 (51.6) |
| Yes | 482 (48.4) |
| 4 to 6 hours | 267 (26.8) |
| In your opinion, in one day, what is the optimal time to work in the current situation? | |
| Less than 4 hours | 79 (7.9) |
| 6 to 8 hours | 493 (49.5) |
| 8 to 10 hours | 124 (12.4) |
| More than 10 hours | 33 (3.3) |
| Do you have the opportunity of time to rest (physically or mentally) during the handling of this COVID-19 outbreak? | |
| No | 263 (26.4) |
| Yes | 733 (73.6) |
| Are you satisfied with the hospital's support for all of you during the handling of this COVID-19 outbreak? | |
| No | 130 (13.1) |
| Yes | 866 (86.9) |
| Do you need psychological help from the Counseling Unit / Department of Psychiatry and Mental Health HTAR? | |
| No | 955 (95.9) |
| Yes | 41 (4.1) |

those who have an adequate amount of rest (AOR = 1.546; 95% CI: 1.106–2.160, p = 0.011). Staff who were not satisfied with the hospital's support were 2.5 times more likely to have depression as compared to those who were satisfied with the hospital during the COVID-19 pandemic (AOR = 2.483; 95% CI: 1.649–3.739, p<0.05). Staff who did not require psychological intervention provided by the hospital were less likely to have depression as compared to those who need of psychological support (AOR = 0.172; 95% CI: 0.112–0.264, p<0.05).

## Anxiety

For anxiety, 71.9% (716) were normal, 7.1% (71) were mild, 12.6% (125) were moderate, and 8.4% (84) were severe Therefore 71.9% (716) were normal and 28.1% (280) had anxiety

**Table 4. Multiple logistic regression between sociodemographic, psychological characteristics and support with depression, anxiety and stress among healthcare workers (N = 996).**

| Variables | | Depression Outcome Normal | Depression Outcome Depression | Depression Adjusted OR | Depression 95% CI Lower | Depression 95% CI Upper | Depression p value | Anxiety Outcome Normal | Anxiety Outcome Anxiety | Anxiety Adjusted OR | Anxiety 95% CI Lower | Anxiety 95% CI Upper | Anxiety p value | Stress Outcome Normal | Stress Outcome Stress | Stress Adjusted OR | Stress 95% CI Lower | Stress 95% CI Upper | Stress p value |
|---|---|---|---|---|---|---|---|---|---|---|---|---|---|---|---|---|---|---|---|
| Age | < 40 years | 611 | 188 | | | | | 560 | 239 | | | | | 575 | 224 | | | | |
| | ≥ 40 years | 164 | 33 | 1.637 | 1.106 | 2.423 | **0.014** | 156 | 41 | 1.632 | 1.141 | 2.334 | **0.007** | 152 | 45 | 1.404 | 0.983 | 2.007 | 0.062 |
| Involvement with COVID-19 patients | No | 270 | 50 | | | | 0.011 | 244 | 76 | | | | **0.044** | 257 | 63 | | | | 0.003 |
| | Direct | 202 | 73 | 0.630 | 0.427 | 0.928 | **0.019** | 200 | 75 | 0.706 | 0.503 | 0.990 | **0.044** | 190 | 85 | 0.596 | 0.418 | 0.849 | **0.004** |
| | Indirect | 303 | 98 | 1.181 | 0.820 | 1.703 | 0.372 | 272 | 129 | 0.812 | 0.574 | 1.149 | 0.240 | 280 | 121 | 1.129 | 0.803 | 1.589 | 0.484 |
| Are you worried that you will be infected with COVID-19? | No | 78 | 26 | | | | | 72 | 32 | | | | | 70 | 34 | | | | |
| | Yes | 697 | 195 | 1.295 | 0.780 | 2.148 | 0.317 | 644 | 248 | 1.249 | 0.784 | 1.990 | 0.349 | 657 | 235 | 1.963 | 1.173 | 3.284 | **0.010** |
| Are you afraid that you could infect your family? | No | 47 | 7 | | | | | 43 | 11 | | | | | 45 | 9 | | | | |
| | Yes | 728 | 214 | 0.538 | 0.216 | 1.341 | 0.183 | 673 | 269 | 0.674 | 0.311 | 1.463 | 0.319 | 682 | 260 | 0.479 | 0.211 | 1.087 | 0.078 |
| Are you worried about the lack of PPE (Personal Protective Equipment)? | No | 77 | 16 | | | | | 72 | 21 | | | | | 75 | 18 | | | | |
| | Yes | 698 | 205 | 0.99 | 0.531 | 1.847 | 0.976 | 644 | 259 | 0.965 | 0.553 | 1.685 | 0.901 | 652 | 251 | 0.802 | 0.447 | 1.438 | 0.459 |
| Do you feel less confident to treat critically ill patients? | No | 424 | 90 | | | | | 393 | 121 | | | | | 400 | 114 | | | | |
| | Yes | 351 | 131 | 0.657 | 0.48 | 0.901 | **0.009** | 323 | 159 | 0.72 | 0.542 | 0.958 | **0.024** | 327 | 155 | 0.638 | 0.476 | 0.856 | **0.003** |
| In your opinion, in one day, what is the optimal time to work in the current situation? | less than 4 hours | 58 | 21 | 0.698 | 0.329 | 1.482 | 0.350 | 56 | 23 | 1.204 | 0.600 | 2.417 | 0.601 | 58 | 21 | 1.265 | 0.618 | 2.587 | 0.520 |
| | 4 to 6 hours | 200 | 67 | | | | 0.777 | 183 | 84 | | | | 0.652 | 192 | 75 | | | | 0.970 |
| | 6 to 8 hours | 395 | 98 | 0.867 | 0.457 | 1.645 | 0.662 | 368 | 125 | 1.118 | 0.603 | 2.074 | 0.724 | 364 | 129 | 1.240 | 0.658 | 2.334 | 0.506 |
| | 8 to 10 hours | 102 | 22 | 0.749 | 0.401 | 1.399 | 0.365 | 88 | 36 | 0.897 | 0.491 | 1.638 | 0.723 | 92 | 32 | 1.188 | 0.641 | 2.202 | 0.584 |
| | more than 10 hours | 20 | 13 | 0.958 | 0.440 | 2.084 | 0.913 | 21 | 12 | 0.996 | 0.470 | 2.111 | 0.992 | 21 | 12 | 1.153 | 0.533 | 2.494 | 0.717 |
| Do you have the opportunity of time to rest (physically or mentally) during the handling of this COVID-19 outbreak? | No | 183 | 80 | | | | | 163 | 100 | | | | | 168 | 95 | | | | |
| | Yes | 592 | 141 | 1.546 | 1.106 | 2.160 | **0.011** | 553 | 180 | 1.677 | 1.233 | 2.281 | **0.001** | 559 | 174 | 1.714 | 1.255 | 2.340 | **0.001** |
| Are you satisfied with the hospital's support for all of you during the handling of this COVID-19 outbreak? | No | 78 | 52 | | | | | 77 | 53 | | | | **0.003** | | | | | | |
| | Yes | 697 | 169 | 2.483 | 1.649 | 3.739 | **0.000** | 639 | 227 | 1.831 | 1.235 | 2.715 | **0.003** | | | | | | |
| Do you need psychological help from the Counseling Unit/ Department of Psychiatry and Mental Health HTAR? | No | 760 | 195 | | | | | 702 | 253 | | | | | 714 | 241 | | | | |
| | Yes | 15 | 26 | 0.172 | 0.112 | 0.264 | **0.000** | 14 | 27 | 0.224 | 0.154 | 0.325 | **0.000** | 13 | 28 | 0.297 | 0.202 | 0.437 | **0.000** |

(Table 2). This study showed that staff aged below 40 years old were twice more likely to have anxiety as compared to those aged 40 and above (AOR = 1.632; 95% CI: 1.141–2.334, p = 0.007) (Table 4). Staff with no involvement with COVID-19 patients were less likely to have anxiety (p = 0.118) as compared to those who had direct involvement with COVID-19 patients (AOR = 0.706; 95% CI: 0.503–0.990, p = 0.044). Staff who were confident in treating critical patients were less likely to have anxiety as compared to those who were less confident in treating critical patients (AOR = 0.720; 95% CI: 0.542–0.958, p = 0.024). Staff who have no time to rest during the COVID-19 pandemic are twice more likely to have anxiety as compared to those who have an adequate amount of rest (AOR = 1.677; 95% CI: 1.233–2.281, p = 0.001). Staff who were not satisfied with the hospital's support were twice more likely to have anxiety as compared to those who are satisfied with the hospital during the COVID-19 pandemic (AOR = 1.831, 95% CI 1.235–2.715, p = 0.003). Staff who did not require psychological intervention provided by the hospital were less likely to have anxiety as compared to those who need of psychological support (AOR = 0.224; 95% CI: 0.154–0.325, p<0.05).

## Stress

For stress, majority, 73.0% (727) were normal, 18.7% (186) were mild, 6.1% (61) were moderate and 2.2% (22) were severe. Therefore, 73% (727) were normal and 27% (269) had stress. (Table 2). Among the staff, 320 staff (32.1%) had no contact with COVID-19 patients, 275 staff (27.6%) had direct contact with COVID-19 patients and 401 staff (40.3%) had indirect contact with COVID-19 patients (Table 3). Staff are worried to get infected with COVID-19 were twice more likely to have stress as compared to those who are not worried to get infected with COVID-19 (AOR = 1.963; 95% CI: 1.173–3.284, P = 0.010) (Table 4). Staff who were confident in treating critical patients were less likely to have anxiety as compared to those who were less confident in treating critical patients (AOR = 0.638; 95%CI: 0.476–0.856, p = 0.003) (Table 4).

## 4. Discussion

The present study examined the mental health toll of the COVID-19 pandemic on HCWs at the government hospital in Klang Valley, Malaysia. To our knowledge, this study is among the first to determine the prevalence and associated factors of depression, anxiety, and stress among HCWs in the government hospitals in Malaysia during the pandemic. The study was conducted during the first and second wave of the pandemic in Malaysia beginning of January 25 and February 27 2020 onwards [43]. The overall prevalence of depression, anxiety, and stress were 22.2%, 28.1%, and 27.0% respectively. These findings were consistent with another study reported by Woon LS et al., (2020) among university HCWs in Malaysia showed almost similar prevalence of depression (21.8%), anxiety (31.6%) and stress (29.1%) [44]. A review by Salari N et al., (2020) showed a high prevalence of psychological impacts with depression (24.3%; 95% CI 18.2–31.6%), anxiety (25.8%; 95% CI 20.5%-31.9%) and anxiety (45%; 95% CI 24.3–67.5%) among HCWs who treating COVID-19 patients [45]. A study by Woon et al., (2020) suggested that the psychological impact of the pandemic among HCWs persist even though the movement control order is lifted [44, 46].

## Depression

This study showed a majority of HCWs (89.6%) were concerned with their worries about the risk of being infected with COVID-19 while managing patients with COVID-19 and due to the lack of PPE supply to the HCWs (90.7%) as a result of this unprecedented pandemic. The finding from this study showed that HCWs were stressed by the fear of exposure to getting infected by COVID-19 patients due to the increasing trend of COVID-19 cases. A study by

Joob B et al., (2020) reported a staff nurse contracted COVID-19 during managing a dengue patient in a Thai hospital although she addressed the precautionary measures during the pandemic COVID-19 [47]. Another study in Wuhan, China showed a large cluster of pneumonia patients and health professionals contracted COVID-19 in the same wards [48]. Being elderly is the greatest risk factor for mortality or morbidity of COVID-19 reported by Wu et al., 2020 who studied hospitalised patients in China [49]. Socio-demographic and underlying medical factors contribute to the risk of psychological disorders in the elderly during the pandemic [50]. Similar findings of higher psychological distress among older Jordanian HCWs during the COVID-19 pandemic were studied by Alnazly E et al., (2021) [51]. Working directly with Covid-19 patients was related overall to greater psychological distress than working indirectly. Ge et al., (2021) reported that transmission of COVID-19 among HCWs were common with index patient although they were clinically present without symptoms due to long-period exposure to close contacts [52]. This present study showed a positive association between depression and being a young HCWs (aged below 40 years old), namely fear of exposure to COVID-19 infection and being infected when managing COVID-19 patients during the pandemic. Furthermore, heavy workload and lack of coping with difficulties during the pandemic warrant social or psychological support from psychological services at the workplace.

## Anxiety

Our current study also showed that there is an association between inadequate rest from long hours of working shifts during the pandemic COVID-19 with psychological distress among HCWs. In a study by Subhas N et al., (2021), shift work consistently predicted anxiety among frontline healthcare workers [53]. These findings were replicated in another study, whereby shift work consistently predicted anxiety amongst frontline healthcare workers [54]. A study by Luceño-Moreno et al., (2020) suggested that HCWs who take a break from long working hours showed lower levels of stress [54]. de Sire A et al., (2021) did a survey among physical therapists related to work and healthcare issues related to COVID-19 showed the importance of a healthy psychosocial work environment to enhance job satisfaction of all health professionals to avoid burnout syndrome and avoid role conflict during the COVID-19 pandemic [55]. Besides breaks from long hours of working shifts, our study showed HCWs need psychological support during COVID-19 pandemic at working place. This can be done by support and empowerment through education and consultation for HCWs to overcome anxiety, stress and depression. Direct support from management can help the HCWs to develop a positive attitude and manage their work-related stress related [50, 56, 57]. HCWs need to feel safe and confident while working at health facilities or working areas where they will be protected at work [58].

## Stress

This present study showed a strong association between stress and feeling worried to get infected with COVID-19 among the staff. As a result, stress may suppress individual's confidence in treating critical patients. In this sense, especially during pandemic situations, majority of our healthcare workers succumb with stress as an immediate effect on them. A previous study by Woon LS et al., (2020) reported fear of exposure to COVID-19 among healthcare workers at university hospitals in Malaysia [44]. Another study done by Alnazly et al., (2021) on Jordanian healthcare workers reported a positive association between fear and stress [51]. A review by Salari N et al., (2020) reported a high prevalence of stress among healthcare workers [21]. These highlighted the need of providing timely psychological support or treatment to

healthcare workers who suffered psychological distress especially during pandemic to ensure the mental health state of healthcare workers is kept at optimal level.

## 5. Limitations

This study has some limitations. Firstly, the respondents recruited were all HCWs from a government hospital without any control group that was not involve at all with Covid-19 patients and this may not be able to generalise the result. Secondly, using psychometric questionnaires that were not designed to evaluate depression, anxiety and stress in the context of the pandemic do not reflect the true scenarios as psychological distress may have been undiagnosed. Thirdly, we didn't conduct pre-test and post-test of DASS-21 after giving out the intervention. So, we couldn't quantify the level of reduction of depression, anxiety and stress among the participants after prescribed the intervention.

Lastly, this current study reported the prevalence of depression, anxiety and stress should be interpreted with caution because all respondents were enrolled in a serial programme on Mental Health and Psychosocial Support in Covid-19 (MHPSS COVID-19). Thus, the findings could be lower than what would be reported among healthcare workers who are not enrolled in a similar programme. A longitudinal study involving all HCWs across all categories would have yielded better information since this study is only cross-sectional. Besides, details of personal and familial issues which could also affect the mental state of respondents were not collected. We didn't consider ethnic group in further analysis since Malay-ethnic represent more than half of the population followed by the Chinese and Indians. We excluded non-malay speakers among HCWs since majority Malaysians is using common language which is the Malay Language benefited using the malay version of DASS-21. Those HCWs didn't give their consent to undergo the survey, they can receive the psychological and emotional support in view of the importance of early mental health treatment. Nevertheless, this study provides unique opportunities for robust evaluation of mental health status among HCWs.

## 6. Conclusions

Our findings revealed the significant risk of adverse mental health outcomes for HCWs treating COVID-19 patients in hospital settings during the peak first and second wave of pandemic in Malaysia. Further research is important to understand the extent of the psychological impacts of pandemic on HCWs over time and identify effective psychological supports to meet their psychological needs and reduce the risk of long-term psychological damage in this COVID-19 pandemic and beyond.

## Supporting information

**S1 Appendix. Malay version of DASS-21.**
(TIFF)

## Acknowledgments

The authors would like to thank the Director General of Health Malaysia for the permission to publish this paper. We also thank all psychiatric Hospital Tengku Ampuan Rahimah Klang, Selangor and staff from all tertiary hospitals in Klang Valley for their continuous support.

## Author Contributions

**Conceptualization:** Nor Asiah Muhamad, Natasha Subhas, Norni Abdullah.

**Formal analysis:** Nor Asiah Muhamad, Normi Mustapha.

**Investigation:** Fatin Athira Tahir.

**Methodology:** Nor Asiah Muhamad.

**Project administration:** Natasha Subhas, Norni Abdullah.

**Resources:** Natasha Subhas.

**Supervision:** Nor Asiah Muhamad, Natasha Subhas, Norni Abdullah, Rimah Melati AB Ghani.

**Validation:** Nor Asiah Muhamad, Normi Mustapha.

**Visualization:** Nor Asiah Muhamad.

**Writing – original draft:** Nor Asiah Muhamad, Natasha Subhas, Norni Abdullah, Muhammad Arif Muhamad Rasat, Rimah Melati AB Ghani, Fatin Athira Tahir, Anne Nik Ismaliza Ishak, Vevehkanandar Sivasubramaniam, Alinazarine Hassan, William Wei Liang Goh, Kok Liang Teng, Ainul Izzah Abdul Manan, Rosmawati Mokhtar, Amrit Kaur Baljit Singh, Kher Shean Ng.

**Writing – review & editing:** Nor Asiah Muhamad, Natasha Subhas, Norni Abdullah, Muhammad Arif Muhamad Rasat, Rimah Melati AB Ghani, Anne Nik Ismaliza Ishak, Vevehkanandar Sivasubramaniam, Alinazarine Hassan, William Wei Liang Goh, Kok Liang Teng, Ainul Izzah Abdul Manan, Rosmawati Mokhtar, Amrit Kaur Baljit Singh, Kher Shean Ng.

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
