## [Decision Letter · Decision Letter 0]

10 Oct 2022

PGPH-D-22-00535

METER (Mental health Emergency Response) Program: Psychological Impact of the COVID-19 Pandemic on healthcare workers in a Tertiary Hospital in Klang Valley, Malaysia

Dear Dr. MUHAMAD,

Thank you for submitting your manuscript to PLOS Global Public Health. After careful consideration, we feel that it has merit but does not fully meet PLOS Global Public Health’s publication criteria as it currently stands. Therefore, we invite you to submit a revised version of the manuscript that addresses the points raised during the review process.

Please ensure native English speaker ( and English editor) proofread the revised version of this paper to improve the readability and clarity of the message being communicated.

We look forward to receiving your revised manuscript.

Kind regards,

Joel Msafiri Francis, MD, MS, PhD

Academic Editor

Journal Requirements:

Additional Editor Comments (if provided):

Please ensure native English speaker ( an English editor) proofread the revised version of this paper to improve the readability and clarity of the message being communicated.

Reviewers' comments:

Reviewer's Responses to Questions

**Comments to the Author**

1. Does this manuscript meet PLOS Global Public Health’s publication criteria? Is the manuscript technically sound, and do the data support the conclusions? The manuscript must describe methodologically and ethically rigorous research with conclusions that are appropriately drawn based on the data presented.

Reviewer #1: Yes

Reviewer #2: Yes

Reviewer #3: Yes

Reviewer #4: Yes

2. Has the statistical analysis been performed appropriately and rigorously?

Reviewer #1: Yes

Reviewer #2: Yes

Reviewer #3: Yes

Reviewer #4: Yes

3. Have the authors made all data underlying the findings in their manuscript fully available (please refer to the Data Availability Statement at the start of the manuscript PDF file)?

Reviewer #1: Yes

Reviewer #2: Yes

Reviewer #3: Yes

Reviewer #4: No

4. Is the manuscript presented in an intelligible fashion and written in standard English?

Reviewer #1: Yes

Reviewer #2: No

Reviewer #3: Yes

Reviewer #4: No

5. Review Comments to the Author

Reviewer #1: Dear Authors,

The data presented was very extensive and could be simplified but my only fear is the meaning could be lost if the simplification was not done intelligibly. Having said that, the wealth of information does reflect on the targeted issue. However the limitation is rather concerning. The design itself is notable sufficient without the mention of a bigger population that is not part of the actual study despite it could give a better reflection on the issue. But this is not the issue within the manuscript but rather to be pondered by the researcher.

Reviewer #2: Thanks for the opportunity to review this manuscript. This is an important topic that will allow policy makers and healthcare institutions to better support their staff, especially in outbreak situations. I have a few major comments below:

In general, the manuscript also requires further proofreading for grammar - I have not provided specific recommendations as there are errors throughout the manuscript. Given that PLOS Global Public Health does not copyedit accepted manuscripts, this would be a necessary major revision made by the authors. The team may consider getting a co-author to proofread the paper once again, or use software like Grammarly to assess the manuscript and create a revision. My comments below therefore reflect suggestions in terms of content, but not writing style.

ABSTRACT

Lines 35-37: This statement should also be accompanied by the corresponding statistical output (adjusted odds ratio etc.)

INTRODUCTION

Line 47: To be more specific, it was declared by the WHO as a public health emergency of international concern (PHEIC)

Lines 60-62: This is vague - the authors could clarify what 'developing a lack of resilience' means because resilience is a psychological construct that should be described and further substantiated. Furthermore, are there specific 'various conditions' that the authors are referring to?

Lines 62-64: This sentence also seems vague - is this referring to any data in particular?

Lines 68-70: This doesn't seem relevant to the study given that the focus is on healthcare workers - unless the authors are making a link between HCWs as being at greatest risk for infection too and therefore being patients themselves?

Lines 73-75: For these 'common disorders associated with the outbreak' - among whom would the authors be referring to? These survivors mentioned above?

Lines 75-76: Same for here - among whom are these psychological sequelae present?

Lines 76+: In general, this feels like a separate point - the data may refer more to the impact of the pandemic (movement control measures + everything else) rather than clinical co-morbidities. I feel like there is a conflation of these issues in the second paragraph and the authors should make an effort to separate them for clarity.

Line 101: COVID-19 as a disease in patients? In other words, are the authors saying that getting infected with COVID-19 and dealing with it is a huge stressor in patients?

METHODS

Line 146: What would a 'serial' programme be referring to? Please do clarify this as it seems slightly vague

Lines 191-193: Would this mean that variables with p<0.05 at bivariable level were included in the multivariable analysis? The p-value at which statistical significance is taken at should be specified

Otherwise, the methods are well-described

RESULTS

Table 4: Given that the tables should be understood even when read without the main text, terms such as 'Involvement' could be expanded a little as it's hard to tell what this means (e.g. involvement in MPHSS? Or involvement in COVID care?) I know this is evident in the main text but does not work well as a standalone term in the table.

DISCUSSION

Limitations: Given that the survey was done in Malay only, is this reflective of all HCWs in Malaysia? E.g. were the participants in MPHSS ethnically diverse?

Limitations: The authors could also discuss briefly what this means for HCWs who are not linked to interventions - would it be that those who are surveyed have better outcomes already given that they were recruited in the context of an intervention?

Reviewer #3: The manuscript was able to be understood well and it was written clearly to present the findings. However, I find that the repetitive nature of how the results were written make me confused. The discussion was well written and conclusion of the manuscript do depict the objective of the study.

Reviewer #4: Title of manuscript: METER (Mental health Emergency Response) Program: Psychological Impact of the

COVID-19 Pandemic on healthcare workers in a Tertiary Hospital in Klang Valley,

Malaysia

Thanks for asking me to review the manuscript titled, “METER (Mental health Emergency Response) Program: Psychological Impact of the COVID-19 Pandemic on healthcare workers in a Tertiary Hospital in Klang Valley,Malaysia”

The topic is very topical and I commend the authors for putting this together. Overall, the findings are important and of value to a national and international audience.

I think the paper would benefit from substantial editing by an English language editor to make it easier to understand and provide adequate feedback. There were a number of words that were missing and in some instances it was somewhat difficult understanding the information that was presented.

I also think that it might be beneficial to modify the title so that it aligns better with the results presented. Authors also need to re-arrange their results and discussion in line with the objectives. This will improve the flow and readability of the manuscript.

Specific issues are outlined below

Title

Findings on the psychological status of the Healthcare workers prior to the pandemic were not presents making it difficult to make an assertion that their current psychological status was as a result of the pandemic.

I suggest a rephrasing of the title.

Abstract

Lines 21 – 23: “This study aims to identify the prevalence and factors of depression, anxiety and stress among HCWs in Malaysian healthcare facilities in the midst [of] the pandemic due to the SARS-CoV-2.”

Line 23: The word, “of”, is missing. Please see above and insert.

Lines 28 - 30:

28 Results: Of the1,300 staffs [who] attended the Mental Health and

29 Psychosocial Support in Covid-19 (MHPSS COVID-19) programme, 996 staffs (21.6% male,

30 78.4% female) completed the online survey (response rate: 76.6%). Result showed [that staff] aged] 31 above 40 years old were almost two times higher [please, replace the word “higher” with, “more likely”] to have anxiety (AOR=1.632; 95% CI=1.141-

32 2.334, p:0.007) and depression (AOR=1.637; 95% CI=1.1.06-2.423, p:0.014) as compared to

33 [staff who were] less than 40 years old.

Insert “who” in line 28.

Lines 35 – 37:

35 “… … …HCWs

36 with stress, anxiety and depression showed less confident to treat critically ill patients and

37 need psychological help during outbreak”.

Please revise the statement (lines 35 – 37) above. Statement is not clear.

Conclusion: This study showed the importance of

38 a healthy psychosocial support to reduce psychological distress among HCWs when working

39 or coping during COVID-19 pandemic or outbreak.

Please, revise the statement above (Conclusion, lines 38 and 39). Statement is not clear.

Introduction

Lines 44 and 45: Use a consistent spelling for aetiology.

Line 47: due [to] its rapid spread and declared a pandemic on 12 March (4-6).

Insert missing word, “to” in the statement above.

Lines 59 – 60: Almost one-third of health care workers (HCWs) experiencing high levels of distress (17, 18).

Please change “experiencing”, to “experienced”

Lines 50 – 100: This paragraph is rather long and has more than one idea. I suggest that the paragraph be revised and split into two so each paragraph has one main idea.

Lines 101 to 102: COVID-19 is itself a huge stressor, as there are [change “are” to “were”] no existing medicine or vaccines during the start of the pandemic

Lines 121 to 122:

121 “Information on mental health status is

122 relevant for future programme on targeted group and evidence inform policy”.

Please revise the statement (lines 121 – 122) above. It is not clear.

Lines 134 to 142. Authors are describing the research objectives and hypothesis. The statements are not clear. I suggest the authors revise these objectives for clarity.

The Introduction needs substantial revision.

I suggest that an English editor assist with revising the paragraphs, correcting mistakes and arranging the paragraphs such that each paragraph has one main idea.

The study objective and hypothesis stated in lines 130 to 137 are not exactly in line with the title which is, “Psychologic impact of COVID-19 on healthcare workers in a tertiary hospital in Klang Valley, Malaysia. Authors need to either revise the title or revise the study objectives and hypothesis. I suggest they review these after examining their results so they can decide on the title and objectives.

Lines 130 - 137

130 … … … Therefore, this study

132 aimed to determine the prevalence of depression, anxiety, and stress among HCWs in public

133 hospital in Malaysia. This study hypothesised a relevant prevalence rate for moderate to

134 severe psychological distress among HCWs. This study also aimed to examine the

135 relationships between sociodemographic traits, psychological characteristics and support

136 with depression, anxiety, and stress with the hypothesis that psychological characteristics and

137 support inversely associated with depression, anxiety, and stress.

Methods

Requires revision. Authors may wish to review the Strobe guidelines and revise the methods section accordingly.

Some of the text uses red font, please correct.

Under data collection: The authors largely described the instruments here and not the data collection process.

Lines 165 to 169.

This is a bit unclear. Authors mentioned that they used the modified version of Depression, Anxiety, and Stress Scale−21 (DASS– 21) – line 166. In line 169, they then mention using the BM DASS-21 scale. Is this the same instrument? Please use one consistent and correct term for the scale. It is also important to provide a brief explanation of the items in the scale and how participants are categorized as having depressive symptoms or not.

It isn’t clear what the psychological first aid tool was used for. I assumed it was to provide immediate help to study participants if they had complaints suggestive of psychological distress (as explained in lines 160 – 163). However, in lines 172 – 184, the authors mention that it was used as a screening tool.

Please see the description in the sub-section, “Psychological First Aid Screening Application and Reporting” lines 172 – 184.

Results

Lines 210 – 211

77.8% (775) were normal, 10.4% (104) were mild, 7.1% 210 (71) were moderate, and 4.6% (46)

211 were severe. Therefore, 77.8% (775) were normal and 22.2% (221) had depression.

I think the authors need to provide a brief description of the Malaysian version of the scale if this was what was used and also explain how the items in the scale are categorized. These could be attached as an appendix.

Lines 242 to 285

Findings on Multiple Logistic Regression between sociodemographic, psychological characteristics and support with Depression,

286 Anxiety and Stress

287 among healthcare workers are reported in this section.

Please re-order the report commencing with depression, anxiety and then stress (not stress, anxiety and depression, as it is currently reported). This improves the flow of the writing and reading, as the scale is called a DAS scale and presumably the questions are in this order.

As the participants were not asked about DAS symptoms prior to the pandemic and no results either from the participants’ records or other valid sources were explored and presented, it is not clear whether the prevalence reported was due to the pandemic, thus authors cannot conclude that the symptoms were related to the pandemic.

It might be better to title the current paper, “factors associated with depression, anxiety, and stress among HCWs in public hospital in Malaysia during the COVID-19 pandemic”.

This better reflects the results and is also in line with the study objective, typed in lines 131 to 133

131 Therefore, this study

132 aimed to determine the prevalence of depression, anxiety, and stress among HCWs in public

133 hospital in Malaysia.

This is also in line with the statement the authors made in the Discussion section: Lines 291 to 296.

291To our knowledge, this study is among the

292 first to determine the prevalence and associated factors of depression, anxiety, and stress

293 among HCWs in government hospital in Malaysia during the pandemic”

Lines 297 to 299

297These findings

298 suggest that the psychological impact of the pandemic among HCWs persist even though

299 movement control order has been eased at Malaysia.

The above statement is not clear.

“these findings” …. Which findings are being referred to?

How do the findings, “suggest that the psychological impact of the pandemic among HCWs persist even though 299 movement control order has been eased at Malaysia”?

Lines 302 to 307.

302 The finding from this study showed that HCWs were stressed

303 from the fear of exposure to get infected from COVID-19 patients were significant. This finding

304 is similar to a study from other country that found HCWs have a higher risk of contracted with

305 COVID-19 at workplace. Study by Joob B et al (2020) reported a staff nurse contracted with

306 COVID-19 during managing dengue patient in a Thai hospital (45). Another study in Wuhan,

307 China showed a large cluster of pneumonia patients and health professionals

Please revise the statement above (lines 302 to 307)

Please revise this statement as the meaning is not quite clear: “The finding from this study showed that HCWs were stressed from the fear of exposure to get infected from COVID-19 patients were significant.”

HCWs reporting a fear of getting exposed (line 303), is not the same as health workers actually getting infected. Hence authors’ statement that,

“The finding from this study showed that HCWs were stressed from the fear of exposure to get infected from COVID-19 patients were significant. This finding is similar to a study from other country [which country is this?] that found HCWs have a higher risk of contracted with COVID-19 at workplace.

Authors could say that the health care workers were justified to be anxious about getting COVID-19 from the workplace as studies had reported that healthcare workers got the virus from the workplace.

Lines 309 to 311. The statements in these lines are not clear

309 HCWs above 40 years old in this study showed a significant psychological impact of anxiety

310 and depression. This could be a possible reason of being old workers is that the risk of

311 contracted with COVID-19 infection is higher than others

Lines 335 – 336 – Statement is not clear

“Besides break from long hours of working shifts, our study showed HCWs need psychological support during COVID-19 pandemic at working place”

It isn’t clear how the study showed that, “showed HCWs need psychological support during COVID-19 pandemic at working place”.

Did the authors ask questions about whether or not the participants wanted psychological support at the workplace during the pandemic? I don’t think I saw this result.

Limitations

How would collecting data on effect of depression, anxiety and stress among the general population have improved the current study?

How would conducting a longitudinal study involving all HCWs across all categories which yields better information? What would this information have been useful for?

Conclusion

Lines 356 and 357

“This study showed the importance of a healthy psychosocial support to reduce psychological distress among HWCs when working or coping during COVID-19 pandemic or outbreak”.

I think authors should state that healthy psychosocial support could be useful in reducing psychological distress … … … This is because the study was not an intervention and authors did not have a comparison group of HCWs who did not receive psychological support.

6. PLOS authors have the option to publish the peer review history of their article (what does this mean?). If published, this will include your full peer review and any attached files.

**Do you want your identity to be public for this peer review?** For information about this choice, including consent withdrawal, please see our Privacy Policy.

Reviewer #1: **Yes: **Dr Mohammad Zabri Johari

Reviewer #2: **Yes: **Rayner Kay Jin Tan

Reviewer #3: No

Reviewer #4: No

---

## [Decision Letter · Decision Letter 1]

6 Dec 2022

PGPH-D-22-00535R1

METER (Mental health Emergency Response) Program: Findings of Psychological Impact Status and factors associated with depression, anxiety and stress among healthcare workers in public hospital in Malaysia during the COVID-19 pandemic

Dear Dr. MUHAMAD,

Thank you for submitting your manuscript to PLOS Global Public Health. After careful consideration, we feel that it has merit but does not fully meet PLOS Global Public Health’s publication criteria as it currently stands. Therefore, we invite you to submit a revised version of the manuscript that addresses the points raised during the review process.

We look forward to receiving your revised manuscript.

Kind regards,

Joel Msafiri Francis, MD, MS, PhD

Academic Editor

Journal Requirements:

Additional Editor Comments (if provided):

Thanks for addressing the previous comments and suggestions. In addition to the reviewers comments - I strongly suggest that you proofread the manuscript to improve the flow of ideas and readability.

Reviewers' comments:

Reviewer's Responses to Questions

**Comments to the Author**

1. If the authors have adequately addressed your comments raised in a previous round of review and you feel that this manuscript is now acceptable for publication, you may indicate that here to bypass the “Comments to the Author” section, enter your conflict of interest statement in the “Confidential to Editor” section, and submit your "Accept" recommendation.

Reviewer #1: All comments have been addressed

Reviewer #2: All comments have been addressed

Reviewer #3: All comments have been addressed

Reviewer #4: (No Response)

2. Does this manuscript meet PLOS Global Public Health’s publication criteria? Is the manuscript technically sound, and do the data support the conclusions? The manuscript must describe methodologically and ethically rigorous research with conclusions that are appropriately drawn based on the data presented.

Reviewer #1: Yes

Reviewer #2: Yes

Reviewer #3: Yes

Reviewer #4: Yes

3. Has the statistical analysis been performed appropriately and rigorously?

Reviewer #1: Yes

Reviewer #2: Yes

Reviewer #3: Yes

Reviewer #4: Yes

4. Have the authors made all data underlying the findings in their manuscript fully available (please refer to the Data Availability Statement at the start of the manuscript PDF file)?

Reviewer #1: Yes

Reviewer #2: Yes

Reviewer #3: Yes

Reviewer #4: No

5. Is the manuscript presented in an intelligible fashion and written in standard English?

Reviewer #1: Yes

Reviewer #2: No

Reviewer #3: Yes

Reviewer #4: (No Response)

6. Review Comments to the Author

Reviewer #1: I do not have further comments but the revision done by the author to fulfil the requirements of the second reviewer I deem as sufficiently acceptable. It has also answered my previous concerns better.

Reviewer #2: Thank you for addressing the comments. I believe the manuscript has greater clarity as a result. I'd suggest one more proofread for language as several sentences are not phrased correctly or are unclear. For example, in lines 84-86:

"Healthcare workers not only engage directly or indirectly with the treatment of infected patients but also the risk of being infected themselves, fear of transmission to family members, being stigmatized, rejected and working under stressful pressures."

In this sentence, while grammatically somewhat correct, the style makes it difficult to read and there is lack of clarity. A better phrasing would be:

"Healthcare workers not only engage with the treatment of infected patients, but are also a risk of being infected themselves. Apart from the risk of infection, they face additional stressors, such as the fear of transmitting COVID-19 to family members, being stigmatized or rejected by others based on their occupation, and working under highly stressful conditions."

I would suggest doing another proofread accordingly.

Reviewer #3: It is a good study that is rarely done. However, it will good to capture more data from the professional groups like doctors.

Reviewer #4: Review

The authors have considerably improved the presentation of their findings, however, the manuscript still requires considerable editing especially as the journal will not be editing the manuscript before publishing. It was still rather difficult fully understanding some sections, especially the discussion, conclusion and limitation.

A few examples of statements that require correction are below:

Abstract

HCWs with stress (AOR=0.638; 95% CI of 0.476 - 0.856, p= 0.003), anxiety (AOR=0.720; 95% CI 0.542 - 0.958, p= 0.024) and depression (AOR=0.657; 95% CI 0.480- 0.901, p= 0.009) showed less confident to treat critically ill patients and need psychological help during outbreak.

Please change, “confident”, to “confidence”

Introduction

- The statement below is not clear. Was COVID-19 detected by or among three Chinese nationals? Please cross-check and correct as required.

“In Malaysia, the first case of COVID-19 was detected by three Chinese nationals on 25 January 2020 …”

- Please write out the full meaning of SARS as it appears it is occurring for the first time here.

Line 61: “… … outbreak of the 21st century, SARS”

- Please replace, “the hospital” with some concise details about the hospital e.g., was this a primary care hospital? In which city or country?

“A study conducted in the hospital during the SARS”

- Please rephrase the sentence below:

“Another finding Mak et al.’s paper found that the cumulative incidences of mental health disorders were 58.9% after the SARS outbreak”

- Lines 97 to 101. Please rephrase the sentence below

“Further importance in early detection and treatment of psychiatric disorders during the Covid-19 outbreak was shown in a large, retrospective cohort study involving 62,354 participants in the United States which reported bidirectional associations in terms of the development of psychiatric disorders and acquiring Covid-19 infection (32).

- Authors are encouraged to discuss depression, anxiety and stress in the same order throughout the manuscript for consistence. The title is depression, anxiety and stress, hence it improves readability of this ordering is maintained, as much as possible.

- The introduction is an improvement on the previous version that was submitted, but it still requires editing.

Materials and methods

This is an improvement on the previous version. The current version however still requires editing. Typically, this section begins with a description of the study area and not the ethical issues.

Authors should please review the Strobe guidelines for writing observational studies and other manuscripts published in the journal.

- Lines 172 to 176 are repeated in Lines 184 to 186

“A Psychological First Aid (PFA) form was distributed online through a google form as a

first aid screening which was sent out via QR code and electronic messaging system to all participants in the programme.”

- Lines 204 to 206

Please explain why these variables were included in the adjusted model.

“Age, gender, and job category were predetermined to enter into the adjusted model, regardless of their bivariate association with each outcome”

- Data collection

The sequence is a bit confusing. Were all the study participants provided with a mental health intervention before the study? If yes this needs to be known.

I think that the fact that all participants received a mental health intervention has introduced a bias into the study. The intervention would result in a lower level of depression, anxiety and stress. Authors need to discuss this in their limitations section.

Results

- Please change staffs to staff in the opening sentence and in other sentences where “staffs” is used instead of “staff”.

- Delete the word, “staff”, highlighted below. It is not necessary to have it here

“Of these, 996 staffs completed the online survey which gave a response rate of 76.6%”.

- The tables might need to be formatted in line with the journal guidelines. Please review other published papers for this.

- Line 245: Please change “staffs” to “staff”. Also change this in other sections in the manuscript.

- Lines 245 to 287. Please summarize the information in table 4. Too much detail is currently presented.

- Lines 299 and 300

Please delete, “for” in the sentence below

“A review by Salari N et al (2020) showed a high prevalence of psychological impacts among HCWs who treat for COVID-19 patients (44).”

- Lines 306 to 308

Please revise the statement below. It is currently difficult to understand and not clear what point it is supporting or refuting.

“A study by Joob B et al (2020) reported a staff nurse contracted with COVID-19 during managing a dengue patient in a Thai hospital”

- Lines 311 to 313

Please revise the statement below. The meaning is unclear.

“HCWs above 40 years old in this study showed a high psychological impact of anxiety and depression. This could be a possible reason of being old workers is that the risk of contracting with COVID-19 infection is higher than others (47)”.

- Lines 316 to 319

Statement below is unclear. Please revise

“This is possibly due to the higher risk of contracted COVID-19 infection among index patients with COVID-19. Ge Y et al (2021) reported a greater potential of contracting with COVID-19 patients with quantity of exposure to a patient with COVID-19 among those close contacts (49).”

- Lines 320 to 320

The current study is a cross-sectional study and not able to demonstrate “cause and effect”, hence the claim below can not be made. Authors should please rephrase to indicate the association that they might have observed.

“Our study also showed that HCWs have psychological distress due to inadequate rest from long hours of working shifts during pandemic COVID-19”.

- Lines 323 to 326 is repeated in lines 328 to 331

“… emphasise the importance of maintaining standards in medical centres to foster shorter shifts and ensure enough resting periods to reduce the risk and vulnerability of HCWs against psychological burden and professional burnout in critical care scenarios”.

The discussion, conclusions and limitations require editing to improve understanding by the reader.

Conclusion

- Lines 362 to 366”

The first few sentences (see below) are not clear

“HCWs face considerable challenges during the pandemic. This study showed the

importance of psychosocial and emotional support to reduce psychological distress among HWCs when working or coping during COVID-19 pandemic or outbreak. This is because the study was not an intervention and authors did not have a comparison group of HCWs who did not receive psychological support”.

Authors need to read the manuscript carefully and address all grammatical issues to improve the manuscript.

7. PLOS authors have the option to publish the peer review history of their article (what does this mean?). If published, this will include your full peer review and any attached files.

**Do you want your identity to be public for this peer review?** For information about this choice, including consent withdrawal, please see our Privacy Policy.

Reviewer #1: **Yes: **Dr Mohammad Zabri Johari

Reviewer #2: **Yes: **TAN KAY JIN RAYNER

Reviewer #3: **Yes: **Mohd Fadzli Mohamad Isa

Reviewer #4: No

---

## [Decision Letter · Decision Letter 2]

8 Feb 2023

PGPH-D-22-00535R2

METER (Mental health Emergency Response) Program: Findings of Psychological Impact Status and factors associated with depression, anxiety and stress among healthcare workers in public hospital in Malaysia during the COVID-19 pandemic

Dear Dr. MUHAMAD,

Thank you for submitting your manuscript to PLOS Global Public Health. After careful consideration, we feel that it has merit but does not fully meet PLOS Global Public Health’s publication criteria as it currently stands. Therefore, we invite you to submit a revised version of the manuscript that addresses the points raised during the review process.

We look forward to receiving your revised manuscript.

Kind regards,

Joel Msafiri Francis, MD, MS, PhD

Academic Editor

Journal Requirements:

Additional Editor Comments (if provided):

It would be helpful to address the final minor issues raised by the reviewers.

Reviewers' comments:

Reviewer's Responses to Questions

**Comments to the Author**

1. If the authors have adequately addressed your comments raised in a previous round of review and you feel that this manuscript is now acceptable for publication, you may indicate that here to bypass the “Comments to the Author” section, enter your conflict of interest statement in the “Confidential to Editor” section, and submit your "Accept" recommendation.

Reviewer #2: All comments have been addressed

Reviewer #4: (No Response)

2. Does this manuscript meet PLOS Global Public Health’s publication criteria? Is the manuscript technically sound, and do the data support the conclusions? The manuscript must describe methodologically and ethically rigorous research with conclusions that are appropriately drawn based on the data presented.

Reviewer #2: Yes

Reviewer #4: Yes

3. Has the statistical analysis been performed appropriately and rigorously?

Reviewer #2: Yes

Reviewer #4: Yes

4. Have the authors made all data underlying the findings in their manuscript fully available (please refer to the Data Availability Statement at the start of the manuscript PDF file)?

Reviewer #2: Yes

Reviewer #4: Yes

5. Is the manuscript presented in an intelligible fashion and written in standard English?

Reviewer #2: No

Reviewer #4: Yes

6. Review Comments to the Author

Reviewer #2: Thanks for the revisions.

While the authors have mentioned that a proofread had been completed in their response, I am unfortunately still finding grammatical errors in the writing. For example:

"The first confirmed case of COVID-19 in Malaysia was detected among three Chinese nationality on 25 January 2020 who previously had close contact with person infected with COVID-19 in Singapore."

The phrase "three Chinese nationality" has no noun in it, and therefore is an incorrect sentence.

I believe that the authors need to engage an external proofreader to correct these errors, as these errors have persisted in the past iterations/revisions.

Reviewer #4: This is a greatly improved version. There are very few minor edits for which I have provided suggestions in the uploaded copy of the manuscript. The statements requiring editing are highlighted in blue to make it easy for authors to implement the very minor corrections.

Once these corrections are made, I recommend accepting the publication

7. PLOS authors have the option to publish the peer review history of their article (what does this mean?). If published, this will include your full peer review and any attached files.

**Do you want your identity to be public for this peer review?** For information about this choice, including consent withdrawal, please see our Privacy Policy.

Reviewer #2: **Yes: **Tan Kay Jin Rayner

Reviewer #4: No

---

## [Decision Letter · Decision Letter 3]

22 Mar 2023

METER (Mental health Emergency Response) Program: Findings of Psychological Impact Status and factors associated with depression, anxiety and stress among healthcare workers in public hospital in Malaysia during the COVID-19 pandemic

PGPH-D-22-00535R3

Dear Dr. MUHAMAD,

We are pleased to inform you that your manuscript 'METER (Mental health Emergency Response) Program: Findings of Psychological Impact Status and factors associated with depression, anxiety and stress among healthcare workers in public hospital in Malaysia during the COVID-19 pandemic' has been provisionally accepted for publication in PLOS Global Public Health.

Best regards,

Joel Msafiri Francis, MD, MS, PhD

Academic Editor

Reviewer Comments (if any, and for reference):

Reviewer's Responses to Questions

**Comments to the Author**

1. If the authors have adequately addressed your comments raised in a previous round of review and you feel that this manuscript is now acceptable for publication, you may indicate that here to bypass the “Comments to the Author” section, enter your conflict of interest statement in the “Confidential to Editor” section, and submit your "Accept" recommendation.

Reviewer #2: All comments have been addressed

2. Does this manuscript meet PLOS Global Public Health’s publication criteria? Is the manuscript technically sound, and do the data support the conclusions? The manuscript must describe methodologically and ethically rigorous research with conclusions that are appropriately drawn based on the data presented.

Reviewer #2: Yes

3. Has the statistical analysis been performed appropriately and rigorously?

Reviewer #2: Yes

4. Have the authors made all data underlying the findings in their manuscript fully available (please refer to the Data Availability Statement at the start of the manuscript PDF file)?

Reviewer #2: Yes

5. Is the manuscript presented in an intelligible fashion and written in standard English?

Reviewer #2: Yes

6. Review Comments to the Author

Reviewer #2: (No Response)

7. PLOS authors have the option to publish the peer review history of their article (what does this mean?). If published, this will include your full peer review and any attached files.

**Do you want your identity to be public for this peer review?** For information about this choice, including consent withdrawal, please see our Privacy Policy.

Reviewer #2: **Yes: **Rayner Kay Jin Tan
